# SUMOylation of Na$_V$1.2 channels regulates the velocity of backpropagating action potentials in cortical pyramidal neurons

Oron Kotler[1], Yana Khrapunsky[1], Arik Shvartsman[1], Hui Dai[2], Leigh D Plant[3], Steven AN Goldstein[2]*, Ilya Fleidervish[1]*

[1]Department of Physiology and Cell Biology, Faculty of Health Sciences, Ben-Gurion University of the Negev, Beer Sheva, Israel; [2]Departments of Pediatrics and Physiology and Biophysics, University of California, Irvine, Irvine, United States; [3]Department of Pharmaceutical Sciences, Northeastern University, Boston, United States

*For correspondence:
sgoldst2@hs.uci.edu (SANG);
ilya@bgu.ac.il (IF)

Competing interest: The authors declare that no competing interests exist.

## Abstract

Voltage-gated sodium channels located in axon initial segments (AIS) trigger action potentials (AP) and play pivotal roles in the excitability of cortical pyramidal neurons. The differential electrophysiological properties and distributions of Na$_V$1.2 and Na$_V$1.6 channels lead to distinct contributions to AP initiation and propagation. While Na$_V$1.6 at the distal AIS promotes AP initiation and forward propagation, Na$_V$1.2 at the proximal AIS promotes the backpropagation of APs to the soma. Here, we show the small ubiquitin-like modifier (SUMO) pathway modulates Na$^+$ channels at the AIS to increase neuronal gain and the speed of backpropagation. Since SUMO does not affect Na$_V$1.6, these effects were attributed to SUMOylation of Na$_V$1.2. Moreover, SUMO effects were absent in a mouse engineered to express Na$_V$1.2-Lys38Gln channels that lack the site for SUMO linkage. Thus, SUMOylation of Na$_V$1.2 exclusively controls I$_{NaP}$ generation and AP backpropagation, thereby playing a prominent role in synaptic integration and plasticity.

## Editor's evaluation

This fundamental study describes how the specific SUMOylation of Nav1.2 channels regulates neuronal function by slowing action potential backpropagation from the AIS. This compelling evidence breaks new ground in the role of SUMOylation in modulating synaptic plasticity and will be of interest to neuroscientists working on synaptic transmission and modulation of ion channel activity.

## Introduction

In cortical pyramidal cells, as in many central nervous system (CNS) neurons, action potentials (APs) initiate in the axon initial segment (AIS), the proximal part of the axon where the neuronal membrane is not covered with a myelin sheath. The AIS is characterized by a specialized assembly of scaffolding proteins and voltage-gated channels with distinctive biophysical properties (*Bean, 2007*; *Rasband, 2010*). Classically, APs propagate forward from the AIS into the axonal arbor, triggering neurotransmitter release from presynaptic terminals. APs can also propagate backward into the dendrites of cortical pyramidal cells, where they are proposed to play a role in synaptic plasticity by regulating synaptic strength and the coordination of synaptic inputs (*Stuart and Sakmann, 1994*; *Markram*

*et al., 1997*). Both initiation and propagation of APs are critically dependent on the distribution and properties of voltage-gated Na$^+$ (Na$_V$) channels in specific neuronal compartments (*Stuart et al., 1997*; *Hu et al., 2009*; *Baranauskas et al., 2013*). Therefore, identifying signaling pathways that regulate the biophysical properties of neuronal Na$_V$ channels is key to understanding spike generation, propagation, and integration in cortical circuits (*Cantrell et al., 1996*; *Cantrell and Catterall, 2001*; *Bender et al., 2012*; *Kole and Stuart, 2012*; *Yin et al., 2017*).

Central neurons express, to varying degrees, three primary Na$_V$ channels α-subunit isoforms. Na$_V$1.1, Na$_V$1.2, and Na$_V$1.6 are found in mature neurons, while Na$_V$1.3 channels are also found in the developing nervous system (*Goldin et al., 2000*). Each Na$_V$ isoform has a distinct spatiotemporal distribution and is subject to the activity of specific signaling pathways that regulate the biophysical properties and trafficking behavior of the channel. For example, in mature pyramidal neurons, the AP trigger zone, which is located in the distal AIS, contains almost exclusively Na$_V$1.6 channels (*Lorincz and Nusser, 2008*; *Hu et al., 2009*; *Lorincz and Nusser, 2010*; *Tian et al., 2014*). These channels are also present in the nodes of Ranvier (*Caldwell et al., 2000*). In contrast, the proximal portion of the AIS, soma, and dendrites is believed to contain mostly Na$_V$1.2 channels (*Hu et al., 2009*; *Grubb et al., 2011*). While this difference in the distribution has long made it tempting to posit that the initiation and forward propagation of APs are predominantly dependent on Na$_V$1.6 channels and the backpropagation of APs is dependent on the activity of Na$_V$1.2 channels, studies of the biophysical attributes of the two channels have not previously verified this hypothesis.

While heterologous studies show that Na$_V$1.6 channels activate at more negative voltages than other neuronal Na$_V$ isoforms and have a higher propensity to generate non-inactivating currents, differences in the gating behavior of Na$_V$1.2 and Na$_V$1.6 channels appear to be subtle within their native neuronal milieu (*Smith et al., 1998*; *Zhou and Goldin, 2004*; *Rush et al., 2005*; *Chen et al., 2008*). Indeed, using knockout mice, we found that Na$_V$1.6 channels were not required to determine the initiation site for APs within the AIS, for backpropagation into the dendrites, or for the lower activation threshold voltage for APs that is commonly observed in pyramidal neurons (*Katz et al., 2018*).

Although the biophysical differences between native Na$_V$1.2 and Na$_V$1.6 are subtle, each channel isoform is differentially regulated by neuromodulators. Thus, Na$_V$1.6 channels are less sensitive to inhibition by cAMP-dependent protein kinase or protein kinase C-mediated phosphorylation (*Chen et al., 2008*). This finding explains divergent regulation of Na$_V$ channel subtypes following the activation of D1/D5 dopamine or 5-HT$_{1A}$ serotonergic receptors (*Maurice et al., 2001*; *Yin et al., 2017*). We have shown that ion channels are also subject to post-translational regulation by covalent linkage of small ubiquitin-like modifier (SUMO) proteins (*Rajan et al., 2005*; *Plant et al., 2010*; *Plant et al., 2011*; *Plant et al., 2012*; *Plant et al., 2016*; *Xiong et al., 2017*; *Plant et al., 2020*). Three SUMO isoforms (SUMO1–3) are operative in central neurons and can modulate the gating of specific channels following their conjugation to the ε-amino group of specific lysine residues on the intracellular termini or cytoplasmic loops of the channel subunits. Further, we found the enzymes required to activate, mature, and conjugate SUMO reside in the plasma membrane of *Xenopus* oocytes, tissue culture cells, and neurons (*Rajan et al., 2005*; *Plant et al., 2010*; *Plant et al., 2011*; *Plant et al., 2016*). Although SUMOylation is a covalent post-translational modification, it is a dynamic process subject to rapid reversal by the action of the SENP family of sentrin-specific cysteine proteases. Thus far, we have observed that SUMOylation increases excitability either by decreasing potassium flux through K$_V$ and K2P channels or by increasing the activity of Na$_V$ channels and that the opposite functional effects are mediated by the activity of SENPs. Further, we have observed that SUMOylation status varies among channel types at baseline and in response to environmental stimuli such as hypoxia (*Plant et al., 2016*; *Xiong et al., 2017*; *Plant et al., 2020*), and particularly germane to this study, that SUMO has differential effects on the principal Na$_V$ channel isoforms expressed in central neurons. Specifically, we found that SUMOylation modulates the voltage-dependent gating of Na$_V$1.2 channels via linking to lysine 38, but SUMO does not interact with Na$_V$1.6 (*Plant et al., 2016*).

Here, we tested the hypothesis that SUMOylation regulates the excitability of cortical pyramidal neurons. By combining whole-cell recordings with high-speed fluorescence imaging to simultaneously monitor Na$^+$ flux in different subcellular compartments of L5 cortical neurons, we found that SUMO1 increases excitability via a synergistic effect on subthreshold K$^+$ and Na$^+$ conductances. Thus, SUMOylation suppresses the open probability of K$^+$ channels while concurrently increasing Na$^+$ influx via a leftward shift in the steady-state activation of subthreshold persistent Na$^+$ currents. These effects

are absent in a CRISPR-generated mouse that constitutively expresses Na$_V$1.2-Lys38Gln channels, a channel variant that cannot be SUMOylated. Confirming the long-held notion for their roles in the AIS based on distribution, and consistent with our previous report that SUMO regulates Na$_V$1.2 but not Na$_V$1.6 channels, we demonstrate that SUMOylation of Na$_V$1.2 regulates the velocity of backpropagation in cortical pyramidal neurons independent of the speed at which AP propagate forward from the AIS.

## Results

### SUMO1 increases the excitability of layer 5 cortical neurons

Previously, we showed that the effects of SUMOylation and deSUMOylation of neuronal ion channels underlying $I_{DR}$, $I_{Kso}$, and $I_{Na}$ could be assessed by including purified SUMO1 or SENP1 polypeptides, respectively, in the recording pipette solution (*Plant et al., 2011*; *Plant et al., 2012*; *Plant et al., 2016*). To characterize the effects of the SUMO pathway on the excitability of layer 5 cortical pyramidal neurons, we made whole-cell, current-clamp recordings from these cells using pipettes filled with a solution containing SUMO1 or SENP1 polypeptides at 1000 and 250 pmol/l, respectively. We have previously shown that polypeptides at these concentrations produce maximal effects on K$_V$, K$_{2P}$, and Na$_V$ channels in cultured rat hippocampal neurons, cerebellar granule neurons, human ventricular cardiomyocytes derived from iPS cells, and on channels expressed in heterologous cell systems (*Plant et al., 2010*; *Plant et al., 2011*; *Plant et al., 2012*; *Plant et al., 2016*; *Plant et al., 2020*).

Passive neuronal properties and repetitive firing characteristics were assessed by examining the voltage responses to a series of prolonged hyperpolarizing and depolarizing current pulses delivered via the somatic pipette. Because cortical cells are geometrically complex, we compared data obtained 2 min and 35 min after breakthrough into whole-cell configuration to account for slow intracellular dialysis of the polypeptide into the neurons. As SUMO1 diffused into the cell, the frequency of spike firing in response to a given depolarizing suprathreshold current pulse increased (*Figure 1a*). Thus, the mean instantaneous firing frequency in response to a 0.3 nA current injection increased from 17.5 ± 3.6 Hz immediately after the break-in to 29.0 ± 5.4 Hz (n = 8, p=0.036, paired *t*-test) after 35 min of SUMO1 dialysis. In contrast, dialysis with the SENP1-containing solution caused a gradual decrease in the frequency of repetitive firing over time. The mean instantaneous firing frequency in response to a 0.3 nA current pulse decreased from 20.5 ± 3.4 Hz immediately after the break-in to 7.3 ± 3.5 Hz (n = 8, p=0.002, paired *t*-test) after 35 min of SENP1 dialysis.

Examining the voltage responses to small hyperpolarizing current pulses before and following the SUMO1 and SENP1 dialysis revealed that the polypeptides elicited opposite effects on passive neuronal properties (*Figure 1b*). Thus, the apparent input resistance (R$_{in}$), calculated as a ratio of the steady-state amplitude of the voltage deflection to current amplitude, gradually increased when SUMO1 was included in the pipette, from 96.1 ± 12.7 MΩ at a time of break-in to the cell to 122.5 ± 13.7 MΩ (n = 8, p<0.002) at 35 min of recording (*Figure 1c*). In contrast, dialysis of the neurons with SENP1 caused R$_{in}$ to decrease as a function of recording time from 130.0 ± 24.9 MΩ to 80.4 ± 14.6 MΩ (n = 6, p<0.01). In parallel, the membrane time constant ($\tau_m$) obtained by fitting a monoexponential function to the voltage transient following the end of the hyperpolarizing current pulse was increased by SUMO1 application from 16.8 ± 1.9 ms at the time of break-in to 22.8 ± 2.7 ms (n = 8, p<0.01) and shortened by SENP1 application from 17.7 ± 0.7 ms to 11.2 ± 1.3 ms (n = 8, p<0.001). Recording of similar duration with control intracellular solution had no significant effect on R$_{in}$ (113.7 ± 9.3 vs. 112.8 ± 8.9 MΩ, n = 11, p=0.77) and $\tau_m$ (24.3 ± 2.2 vs. 21.2 ± 1.9 ms, n = 11, p=0.09). These findings indicate that in L5 cortical neurons the SUMO pathway regulates potassium channels that determine the passive membrane properties. Furthermore, the relatively high effectiveness of SENP1 suggests that in L5 neurons, as in other cell types (*Rajan et al., 2005*; *Plant et al., 2011*; *Plant et al., 2012*; *Plant et al., 2016*; *Xiong et al., 2017*), a significant fraction of these channels are SUMOylated under control conditions.

The effect of SUMO1 and SENP1 on repetitive firing may reflect the action of the polypeptides on passive neuronal characteristics or their influence on the ion currents underlying spike generation. Theoretical analysis revealed that while the former mechanism should elicit a parallel shift of the frequency–current (F-I) curve to the right or left along the current axis (*Chance et al., 2002*), the latter should alter the neuronal gain, that is, the steepness of the slope of the F-I characteristic.

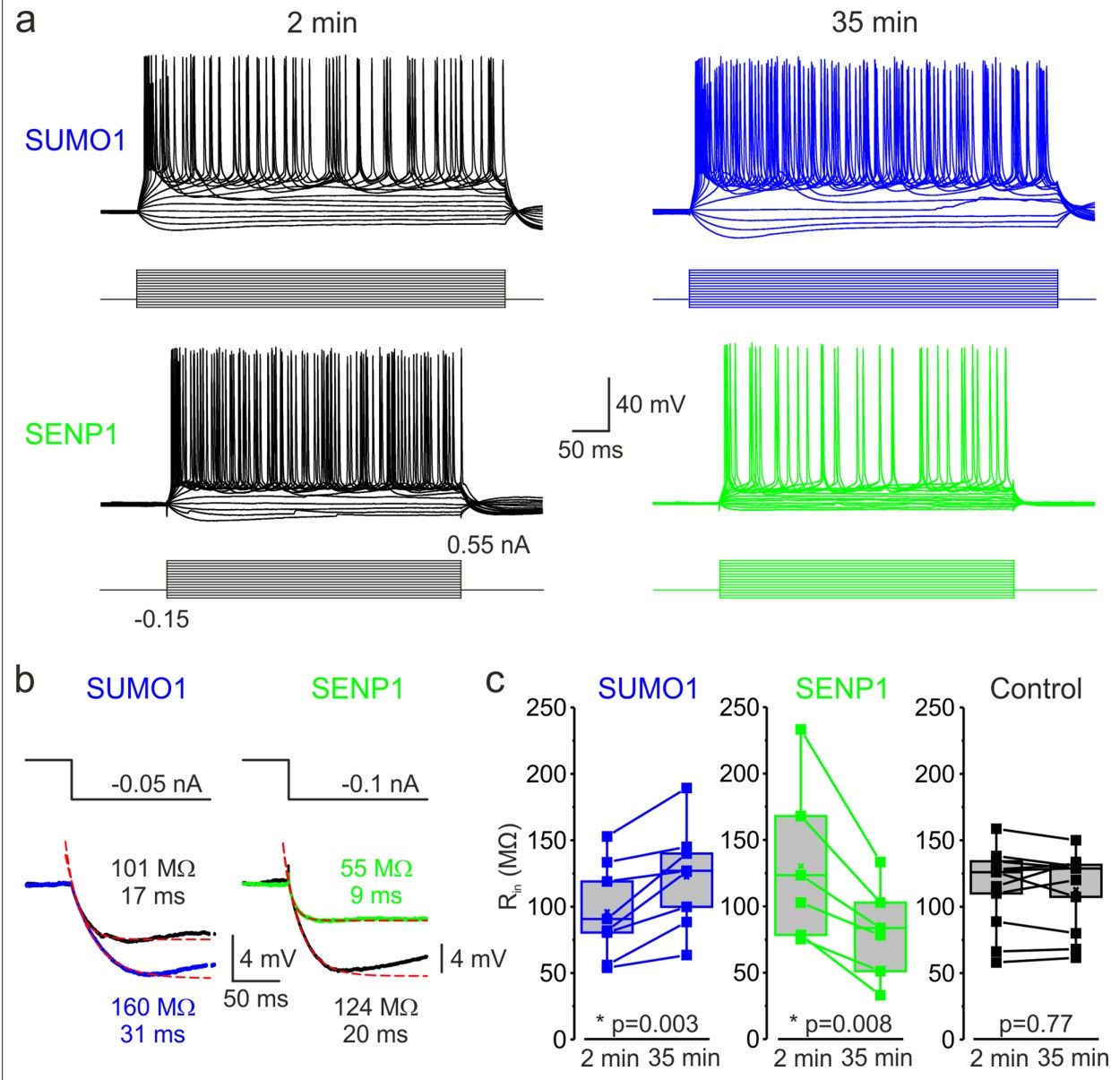

**Figure 1.** SUMOylation and deSUMOylation have opposing effects on the excitability of layer 5 pyramidal neurons. (**a**) Current-clamp, whole-cell recordings from L5 neurons 2 min after the break-in to whole-cell mode (black) and 35 min later demonstrate time-dependent effects of SUMO1 (blue) or SENP1 (green) dialysis on firing frequency. Voltage responses were elicited by injecting 400-ms-long current pulses, which started at –0.15 nA and incremented by 50 pA. (**b**) SUMO1 and SENP1 have opposite effects on passive membrane properties. Voltage responses to a small hyperpolarizing current pulse injection immediately after the break-in (black) and following SUMO1 (blue) or SENP1 (green) dialysis via the whole-cell pipette. Red dashed lines are the best exponential fits of the voltage responses. Notice that the amplitude of voltage deflection and the membrane time constant were enhanced by SUMO1 and decreased by SENP1 dialysis. (**c**) Apparent input resistance ($R_{in}$) increases in SUMO1 dialyzed neurons, whereas it decreases in SENP1 dialyzed cells. The lines connect the paired $R_{in}$ values obtained from the same individual neuron at 2 min and 35 min of recording with SUMO1 (blue), SENP1 (green), and control solution-filled pipette (black). Box plots represent the 25–75% interquartile range of values obtained from neurons dialyzed with SUMO1 (n = 8), SENP1 (n = 6), and control (n = 11) solution; the whiskers expand to the 5–95% range. A horizontal line inside the box represents the median of the distribution, and the mean is represented by a cross symbol (X). p-Values were calculated using Student's *t*-test for paired data.

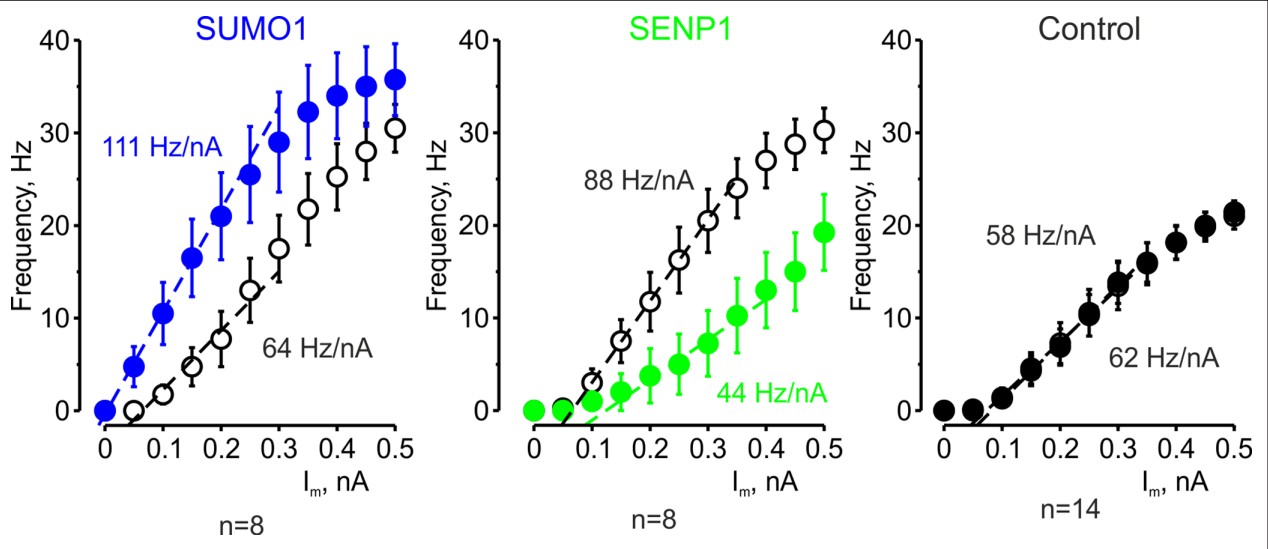

**Figure 2.** The effects of SUMO1 and SENP1 on input/output gain. The frequency–current (F-I) characteristic of L5 pyramidal neurons, constructed by plotting the mean instantaneous spike frequency as a function of depolarizing current pulse amplitude, obtained immediately after the break-in (open black circles) and following 35 min of recording with SUMO1 (n = 8, blue), SENP1 (n = 8, green) and control solution (n = 14, closed black circles) containing pipette. Notice that the F-I curve was shifted to the left and became steeper in SUMO1 dialyzed neurons, whereas in SENP1 dialyzed cells the F-I characteristics were displaced to the right and its slope (dashed line) decreased. The F-I curve showed no significant change in control recordings.

Comparing the linear fits of the mean F-I curves obtained immediately after the break-in and following the SUMO1 dialysis, we found that the curve steepness increased by ~75%, from 64 to 111 Hz/nA (n = 8) (*Figure 2*). Dialysis with control pipette solution had little to no effect on the neuronal gain (62 vs. 58 Hz/nA, respectively, n = 14). In contrast, in recordings with SENP1 containing pipette, the gain decreased from 88 to 44 Hz/nA (n = 8).

To test our hypothesis that SUMOylation of $Na_V1.2$ channels can regulate the excitability of L5 cortical neurons, we used CRISPR/Cas9 to engineer a mouse model carrying $Na_V1.2$-Lys38Gln, a mutation that removes the only SUMO-conjugation site in $Na_V1.2$ channels (*Plant et al., 2016*). The genotype of the mice was verified by PCR screening and sequencing analysis (*Figure 3—figure supplement 1*). Comparison of passive and active electrophysiological characteristics of WT and $Na_V1.2$-Lys38Gln mutant layer 5 pyramidal neurons revealed no significant difference (*Table 1*), indicating that the functional consequences of the mutation are largely compensated. Using whole-cell current-clamp recordings from L5 neurons from the $Na_V1.2$-Lys38Gln mutant mice, we first sought to find out whether SUMO1 and SENP1 dialysis affect the F-I relationship (*Figure 3a*). As in WT neurons, SUMO1 dialysis enhanced the frequency of repetitive firing for a given amplitude of the current pulse, whereas the SENP1 dialysis had the opposite effect. Thus, the mean instantaneous firing frequency in response to a 0.3 nA current injection increased from 18.7 ± 2.6 Hz immediately after the break-in to 24.0 ± 2.4 Hz (n = 6, p=0.013, paired *t*-test) after 35 min of SUMO1 dialysis. In contrast, the mean instantaneous firing frequency decreased from 13.1 ± 1.9 Hz immediately after the break-in to 2.9 ± 1.7 Hz (n = 7, p=0.0005, paired *t*-test) after 35 min of SENP1 dialysis.

Both treatments, however, affected the position of the F-I curve relative to the current axis while little to no effect on its slope was observed (*Figure 3b*), consistent with the hypothesis that, in $Na_V1.2$-Lys38Gln neurons, SUMOylation primarily affects passive neuronal properties. Indeed, in $Na_V1.2$-Lys38Gln mutant neurons, SUMO1 dialysis increased the apparent $R_{in}$ (from 144.2 ± 19.8 MΩ to 171.6 ± 21.1 MΩ, n = 6, p<0.005) whereas dialysis with SENP1 had an opposite effect (from 126.8 ± 19.6 MΩ to 79.8 ± 8.6 MΩ, n = 7, p<0.01) (*Figure 3c*).

**Table 1.** Comparison of electrophysiological characteristics of WT and Na$_V$1.2-Lys38Gln mutant layer 5 pyramidal neurons.

| Parameter | Wild type | Na$_V$1.2-Lys38Gln mutant | Difference |
|---|---|---|---|
| Input resistance (MΩ) | 127.4 ± 11.5 (n = 28) | 134.9 ± 13.6 (n = 13) | NS, p=0.701 |
| Membrane time constant, $\tau_m$ (ms) | 20.4 ± 1.3 (n = 30) | 24.0 ± 2.3 (n = 13) | NS, p=0.149 |
| Voltage threshold (mV)[*] | −57.4 ± 1.4 (n = 17) | −57.5 ± 0.6 (n = 13) | NS, p=0.956 |
| Current threshold (pA)[†] | 448 ± 30 (n = 18) | 514 ± 32 (n = 12) | NS, p=0.151 |
| AP peak (mV) | +36.5 ± 1.4 (n = 17) | +37.1 ± 1.3 (n = 13) | NS, p=0.729 |
| AP dV/dt$_{max}$ (V/s) | 268 ± 25 (n = 17) | 271 ± 20 (n = 13) | NS, p=0.937 |
| AP half-width (ms) | 1.19 ± 0.13 (n = 17) | 1.06 ± 0.10 (n = 12) | NS, p=0.486 |
| F-I characteristics slope (Hz/nA) | 91.0 ± 4.0 (n = 30) | 94.5 ± 7.0 (n = 13) | NS, p=0.653 |

Data are presented as mean ± SE; WT and mutant neurons are compared using the Student's *t*-test for unpaired data.

[*]Voltages were corrected for liquid junction potential of −13 mV (recording temperature of 32°C). Data were collected within 2 min after breaking into the whole-cell configuration.

[†]The current threshold was defined as the minimum amplitude of a 10-ms-long current step that elicited an AP.

## SUMO1 and SENP1 have the opposite effect on the voltage dependence of I$_{NaP}$

In cortical pyramidal neurons, the persistent sodium current operates at a subthreshold range of voltages and is one of the main factors influencing the frequency of repetitive firing, thereby modifying the neuronal gain (*Stuart and Sakmann, 1995*; *Astman et al., 2006*). We have recently shown that in pyramidal cells, most of the whole cell I$_{NaP}$ is generated by somatodendritic Na$^+$ channels (*Fleidervish et al., 2010*; *Shvartsman et al., 2021*). However, because the steady-state activation curve of the AIS channels is shifted to the left by 7–9 mV, most of I$_{NaP}$ at functionally critical subthreshold voltages is axonal. The immunohistochemical evidence indicates that soma, dendrites, and proximal AIS of L5 pyramidal neurons are populated predominantly by the Na$_V$1.2 channels whose activation and inactivation gating is sensitive to SUMOylation (*Hu et al., 2009*; *Grubb et al., 2011*; *Plant et al., 2016*; *Liu et al., 2022*). In contrast, the distal AIS membrane and the Ranvier nodes contain Na$_V$1.6 channels, which are not subject to SUMOylation (*Plant et al., 2016*). To find out how SUMO1 and SENP1 affect the persistent sodium current in different neuronal compartments, we combined whole-cell, voltage-clamp recordings from L5 neurons with high-speed fluorescence imaging of a Na$^+$ sensitive dye, SBFI. A comparison of the voltage ramp-elicited Na$^+$ fluxes revealed that SUMO1 dialysis induces a left shift in the voltage dependence of I$_{NaP}$ activation in soma, proximal apical dendrite, and in the AIS of L5 neurons (*Figure 4a*). Thus, at a voltage of −50 mV, the relatively small fluorescence change in the soma and apical dendrites was significantly increased by SUMO1 dialysis, whereas the amplitude of the Na$^+$ signal in the AIS was less markedly increased (*Figure 4b*).

Measurements of half-activation voltage (V½) revealed that SUMO1 dialysis causes a significant leftward shift in the voltage dependence of activation of both somatic and axonal channels in WT neurons (*Figure 4—figure supplement 1*). However, the application of SUMO1 produced no effect on the voltage dependence of I$_{NaP}$ in neurons from Na$_V$1.2-Lys38Gln mice (*Figure 4—figure supplement 2*). Intracellular application of SENP1 resulted in an opposite effect on the voltage dependence of I$_{NaP}$. Thus, a small but significant rightward shift in the V½ of I$_{NaP}$ was observed in the soma and AIS of neurons from WT but not Na$_V$1.2-Lys38Gln mice (*Figure 5*). These findings indicate that in cortical neurons a portion of the Na$_V$1.2 channels is SUMOylated under control conditions.

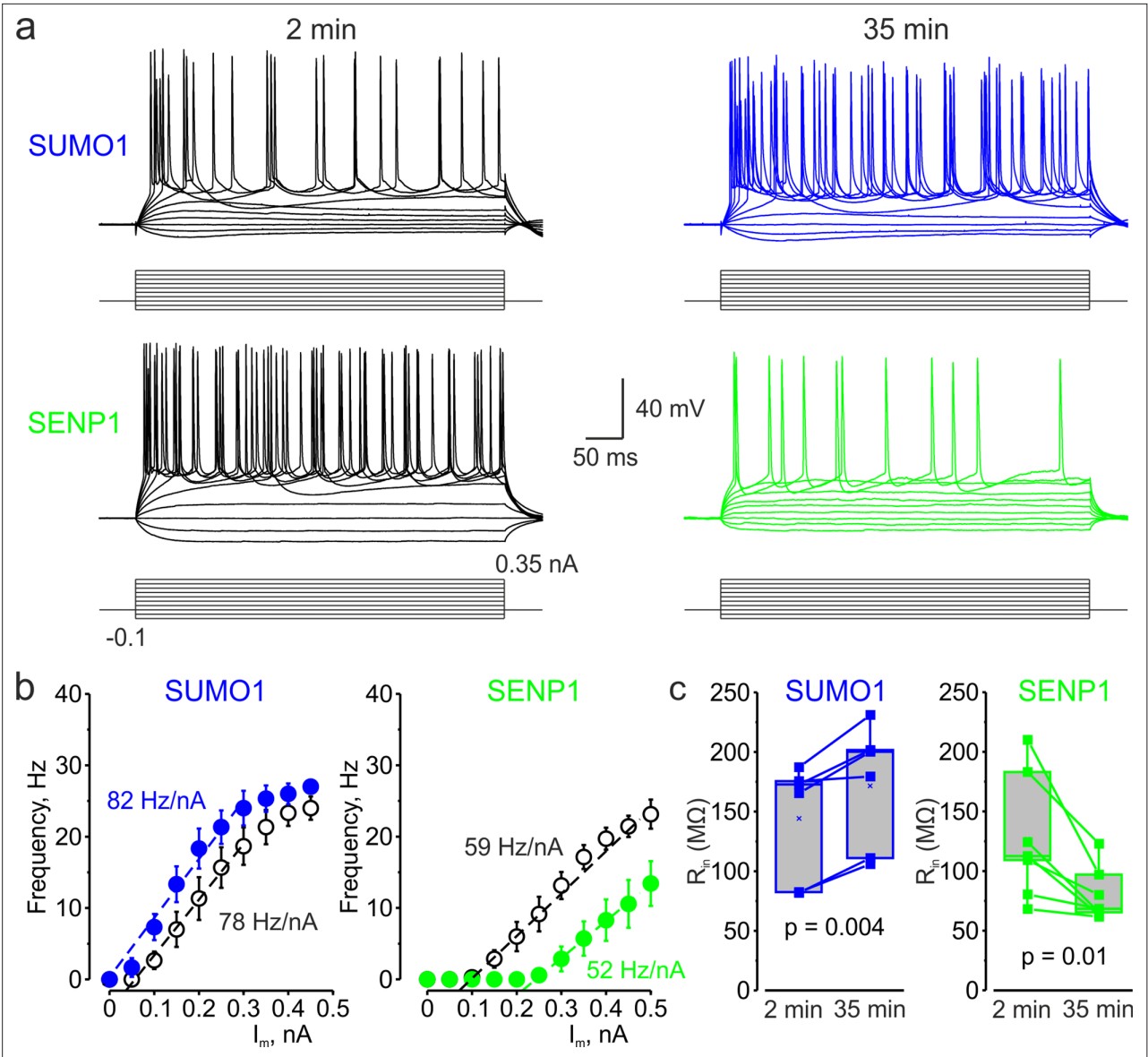

**Figure 3.** In L5 Na$_v$1.2-Lys38Gln mutant neurons, SUMO1 and SENP1 do not affect the gain of the input–output curve. (**a**) Current-clamp, whole-cell recordings from L5 Na$_v$1.2-Lys38Gln mutant neurons immediately after the break-in (black) and following SUMO1 (blue) or SENP1 (green) dialysis. Voltage responses were elicited by injecting 400-ms-long current pulses, which started at –0.15 nA and incremented by 50 pA. (**b**) The F-I characteristic of Na$_v$1.2-Lys38Gln mutant neurons obtained immediately after the break-in (black open circles) and following SUMO1 (n = 6, blue) or SENP1 (n = 7, green) dialysis via the whole-cell pipette. Notice the opposite effects of SUMO1 and SENP1 on the position of the F-I curve over the current axis. Both treatments had little to no effect on the slope of the F-I curve. (**c**) The R$_{in}$ increased over time in SUMO1 dialyzed Na$_v$1.2-Lys38Gln mutant neurons, whereas it decreased in SENP1 dialyzed cells. The lines connect the paired R$_{in}$ values obtained from the same individual neuron at 2 min and 35 min of recording with SUMO1 (blue), SENP1 (green). (**c**) Apparent input resistance (R$_{in}$) increases in SUMO1 dialyzed neurons, whereas it decreases in SENP1 dialyzed cells. The lines connect the paired R$_{in}$ values obtained from the same individual neuron at 2 min and 35 min of recording with SUMO1 (blue) and SENP1 (green) containing solution. Box plots represent the 25–75% interquartile range of values obtained from neurons dialyzed with SUMO1 (n = 6) and SENP1 (n = 7) solution; the whiskers expand to the 5–95% range. A horizontal line inside the box represents the median of the distribution, and the mean is represented by a cross symbol (X). p-Values were calculated using Student's *t*-test for paired data.

The online version of this article includes the following figure supplement(s) for figure 3:

**Figure supplement 1.** Genotyping of transgenic mice obtained through CRISPR-Cas9 targeting of the *Scn2a* gene.

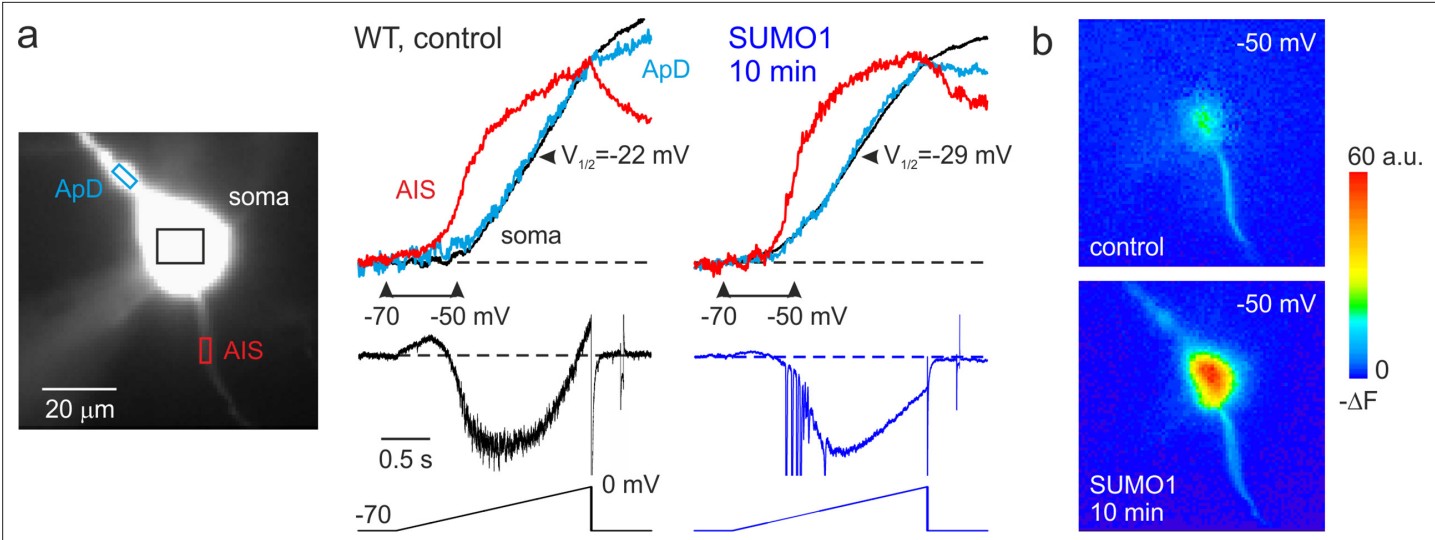

**Figure 4.** SUMO1 causes a leftward shift of $I_{NaP}$ voltage dependence in pyramidal cells from wild-type but not from $Na_v$1.2-Lys38Gln mutant mice. (**a**) Left: WT L5 pyramidal neuron filled with SBFI-containing, $Cs^+$-based solution via a somatic patch pipette, as seen during the fluorescence imaging experiment with a NeuroCCD-SMQ camera. Right: $I_{NaP}$ and normalized somatic (black), axonal (red), and dendritic (cyan) $\Delta F$ transients elicited by 2-s-long voltage ramp from –70 mV to 0 mV immediately after the break-in and following 10 min of dialysis with SUMO1. Notice the leftward shift in voltage dependence of $I_{NaP}$ activation in soma, dendrite, and to a lesser extent, in axon initial segments (AIS). Capacitive and leakage currents were not subtracted. (**b**) Pseudocolor maps of the ramp elicited $\Delta F$ changes between the times marked by the arrowheads in (**a**). Top: voltage ramp from –70 to –50 mV produced $Na^+$ elevation mostly in the AIS. Bottom: following the SUMO1 dialysis, voltage ramp from –70 to –50 mV elicited large $Na^+$ signals also in the soma and dendrites.

The online version of this article includes the following figure supplement(s) for figure 4:

**Figure supplement 1.** $V_{1/2}$ of $I_{NaP}$ activation in the soma (black) and axon initial segments (AIS) (red) of WT pyramidal neurons immediately after the break-in and following 10 min of dialysis with SUMO1.

**Figure supplement 2.** $V_{1/2}$ of $I_{NaP}$ activation in the soma (black) and axon initial segments (AIS) (red) of pyramidal neurons of animals carrying $Na_v$1.2-Lys38Gln mutation, immediately after the break-in, and following 10 min of dialysis with SUMO1.

## SUMOylation of $Na^+$ channels affects voltage-dependent amplification of EPSPs in pyramidal neurons

Changes in the amplitude of $I_{NaP}$ at subthreshold voltages are expected to influence the spatial and temporal summation of synaptic potentials (*Deisz et al., 1991*; *Stuart and Sakmann, 1995*; *Stuart, 1999*). Therefore, we studied the effect of SUMOylation on the amplitude and duration of excitatory postsynaptic potentials (EPSPs) elicited in the pyramidal neuron by brief synaptic stimuli. The EPSPs were measured immediately after break-in to the whole-cell configuration and following 30 min of intracellular dialysis with SUMO1 in WT and $Na_V$1.2-Lys38Gln neurons. SUMO1 did not change the duration of small EPSPs of less than 10 mV in amplitude (*Figure 6a*). In contrast, SUMO1 prolonged the decay time constant of EPSPs greater than 10 mV in amplitude in WT but not $Na_V$1.2-Lys38Gln neurons. In pooled EPSPs obtained from six neurons in each experimental group, SUMO1 dialysis enhanced the steepness of the slope of EPSP integral-to-peak relationship (*Figure 6b*) in WT neurons, whereas SUMOylation had no effect on this relationship for $Na_V$1.2-Lys38Gln cells.

## SUMOylation differentially affects the speed of forward- and back-propagating action potentials

In cortical pyramidal neurons, the $Na_V$1.2 channels are predominantly localized in somatic, dendritic, and proximal AIS membrane, where they are responsible for the propagation of action potentials back into the dendritic tree (*Hu et al., 2009*; *Grubb et al., 2011*). The $Na_V$1.6 channel subtype is present in the distal AIS and in the nodes of Ranvier, and it is responsible for the forward propagation of action potentials into the axonal arbor (*Hu et al., 2009*). Because $Na_V$1.2 and $Na_V$1.6 channels respond differentially to SUMOylation, with the former being susceptible and the latter resistant to SUMO1, we hypothesized that this neuromodulation could differentially affect the speed of forward

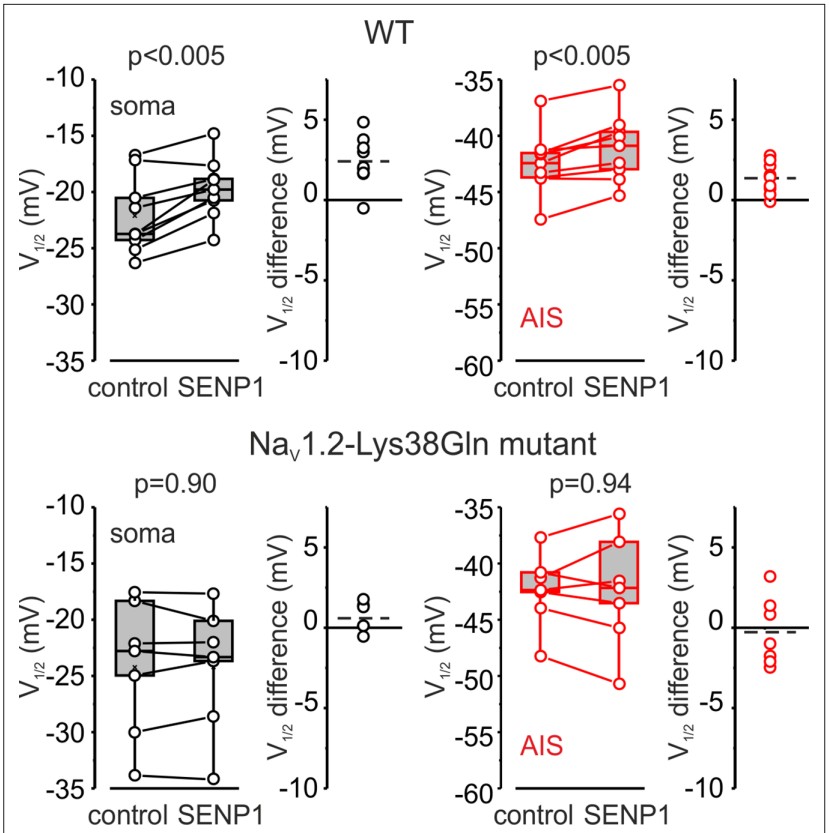

**Figure 5.** SENP1 causes a rightward shift of activation kinetics of $I_{NaP}$ in pyramidal cells from wild-type but not from $Na_v1.2$-Lys38Gln mutant mice. Box plots represent the 25–75% interquartile range of $I_{NaP}$ $V_{1/2}$ activation values in the soma (black), and axon initial segments (AIS) (red) of WT (top, n = 9) and Nav1.2-Lys38Gln mutant (bottom, n = 7) pyramidal neurons immediately after the break-in and following 10 min of dialysis with SENP1; the whiskers expand to the 5–95% range. A horizontal line inside the box represents the median of the distribution, and the mean is represented by a cross symbol (X). p-Values were calculated using Student's t-test for paired data. The lines connect the paired $V_{1/2}$ values obtained from the same individual neuron after the break-in and following 10 min of dialysis with SENP1. $V_{1/2}$ difference plots show a change in $I_{NaP}$ half-activation voltage elicited by SENP1 in individual neurons (open circles); dashed lines show mean $V_{1/2}$ values in WT (n = 9) and Nav1.2-Lys38Gln mutant (n = 7) cells.

and backpropagation of the spikes. Seeking to test this hypothesis directly, we measured the velocity of forward and backpropagation using paired, whole-cell, loose patch recordings to detect the times of the spike arrival from multiple sites along the axo-somatic axis in sequence (*Figure 7*; *Baranauskas et al., 2013*; *Lezmy et al., 2017*). In order to distinguish the axon from other thin processes emerging from the cell body and facilitate the distance measurements between the somatic and axonal pipettes, we filled the neurons for at least 15 min with the Na⁺-sensitive dye SBFI. Because of this and the relatively long time it takes to obtain action currents from multiple axonal locations, we were not able to measure the propagation velocity upon break-in to whole-cell configuration. Thus, we compared the propagation velocities in WT neurons dialyzed with control or SUMO1-containing intracellular solution. As an additional control, the same measurements were taken from the $Na_v1.2$-Lys38Gln neurons dialyzed with SUMO1.

As demonstrated by a representative untreated WT cell (*Figure 7a*), backpropagation velocity (0.10 m/s) was significantly lower than the velocity of forward propagation (0.32 m/s). Dialysis with SUMO1, however, speeded the backpropagation, such that its velocity became almost equal to the speed of forward propagation, ~0.27 vs. 0.23 m/s for forward and backpropagation, respectively (*Figure 7b*). This effect of SUMO1 was not observed in $Na_v1.2$-Lys38Gln neurons, in which the backpropagation was still significantly slower than forward propagation, ~0.26 vs. 0.12 m/s for forward and backpropagation, respectively (*Figure 7c*). Comparison of the ratios of backward and forward

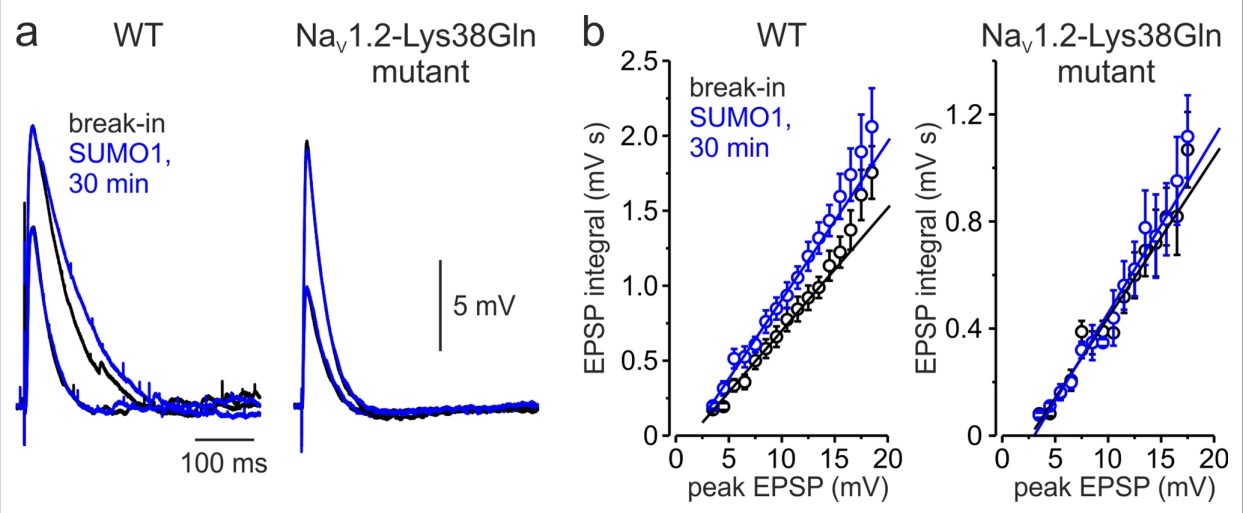

**Figure 6.** Effect of SUMO1 on voltage-dependent amplification of EPSPs in pyramidal neurons from wild-type and Na$_V$1.2-Lys38Gln mutant mice. (**a**) Comparison of small and large EPSPs evoked in WT and Na$_V$1.2-Lys38Gln mutant pyramidal neurons immediately after the break-in (black) and following the SUMO1 dialysis (blue). Notice the slower decay time constant of larger EPSP following SUMO1 dialysis in WT neuron. (**b**) The mean EPSP integral as a function of peak EPSP amplitude after the brake-in (black) and following the SUMO1 dialysis (blue) of the WT (n = 6) and Na$_V$1.2-Lys38Gln mutant (n = 6) pyramidal neurons. Notice the amplification of larger EPSPs in SUMO1 dialyzed WT cells.

propagation velocities revealed a significant increase in WT neurons dialyzed with SUMO1 compared with untreated WT or SUMO1-treated Na$_V$1.2-Lys38Gln cells (*Figure 7d*). To find out whether the leftward shift in voltage dependence of Na$_V$1.2 activation could increase the backpropagation velocity, we studied the dynamics of AP propagation in a simplified compartmental model in which we distributed the Na$_V$1.2 and Na$_V$1.6 channels in accordance with immunohistochemical data (*Hu et al., 2009*). In good agreement with our experimental results, a 6 mV leftward shift in half-activation voltage of Na$_V$1.2 caused an about twofold increase in AP backpropagation velocity (*Figure 7—figure supplement 1*), whereas the forward propagation remained almost unaffected. Thus, our data indicates that in cortical pyramidal neurons SUMOylation of Na$_V$1.2 channels could provide a 'switch' allowing differential regulation of the AP invasion into the dendritic tree and synaptic plasticity, whereas the ongoing neuronal activity that relies on SUMO-resistant, Nav1.6-mediated, spike forward propagation, would not be affected.

## Discussion

We have previously shown that SUMOylation has opposite but synergistic effects on Na$^+$ and K$^+$ channel gating that conspire to increase neuronal excitability. Our present findings in cortical brain slices reveal that SUMO1, on the one hand, increases the inward persistent Na$^+$ current, and on the other hand, decreases the outward potassium current at the subthreshold range of voltages. Together, the SUMOylation of these channels enhances the gain of neuronal responses (*Chance et al., 2002*) to depolarizing current injection by increasing the steepness of the post-spike voltage trajectory towards the next spike threshold. In contrast, we found that deSUMOylation of Na$^+$ and K$^+$ channels by SENP1 decreases neuronal gain, indicating that native neuronal channels are partially SUMOylated under baseline conditions. These findings are congruent with reports describing SUMO-regulation of Na$^+$ and K$^+$ channels in neurons (*Plant et al., 2011*; *Plant et al., 2012*; *Qi et al., 2014*; *Plant et al., 2016*; *Welch et al., 2019*).

We have recently reported that, in Na$_V$1.6-deficient L5 pyramidal neurons, the Na$_V$1.2 channels expressed in the AIS still show a clear hyperpolarizing shift in the voltage dependence of activation compared with somatic channels (*Katz et al., 2018*). One of the goals of this study was to find out whether SUMOylation of Na$^+$ channels (*Plant et al., 2016*) could be, at least partially, responsible for the axo-somatic difference in Na$^+$ channel gating. Because whole-cell recording of transient Na$^+$ current is not achievable in huge, geometrically complex L5 pyramidal neurons (*Spruston et al., 1993*), we used a combination of electrical and Na$^+$ imaging recording to compare the voltage dependence

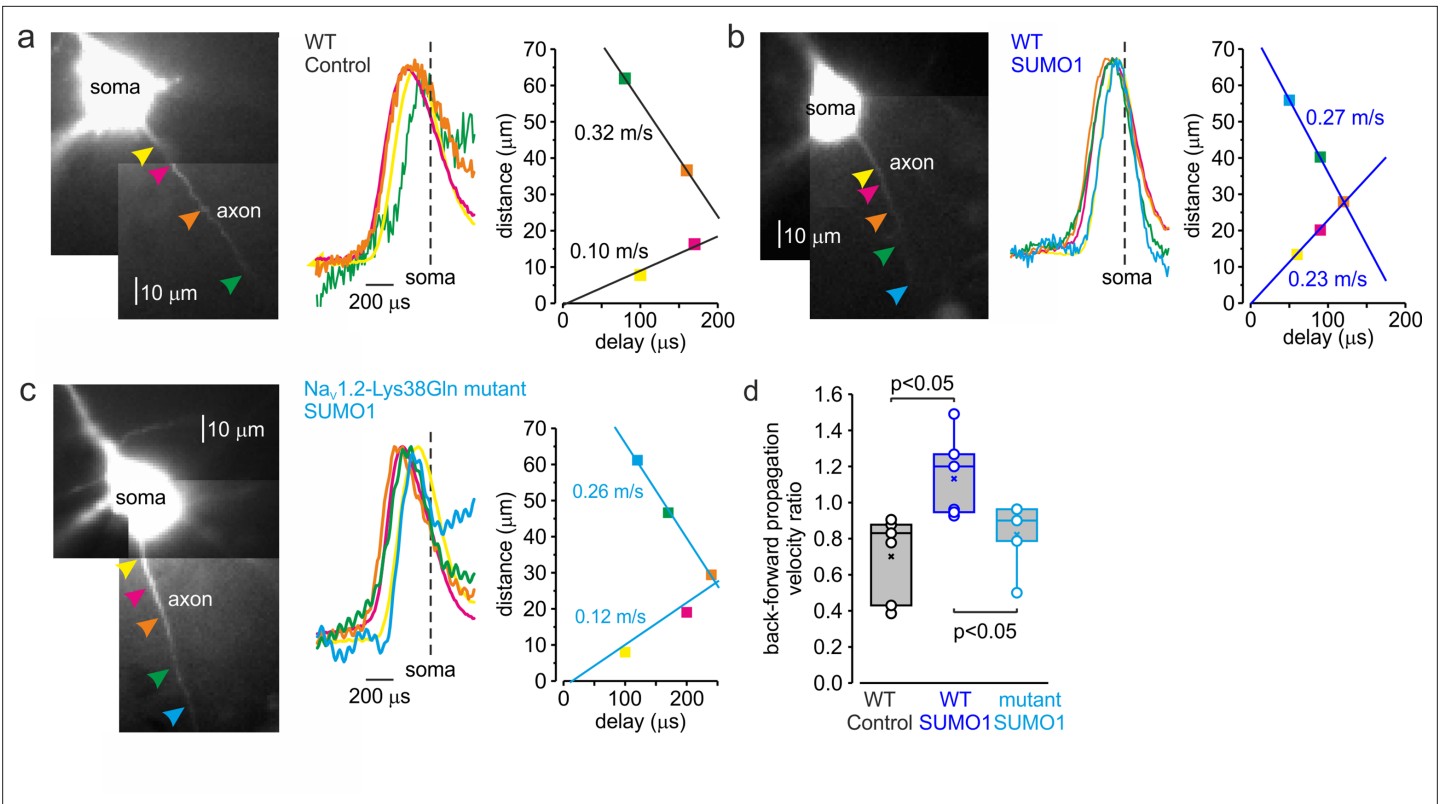

**Figure 7.** SUMOylation differentially affects the velocity of forward- and back-propagating action potentials (APs). (**a**) Left: normalized averaged action currents (n = 500) elicited by a single AP at the axonal regions indicated by arrows to demonstrate the difference in the delay of their onset. The dashed vertical line corresponds to the time of dV/dt$_{max}$ of the somatic action potential. Right: distance from the edge of the soma as a function of delay of spike initiation plotted. Note that AP initiates in a region between the pink and brown arrows and propagates with an apparent conduction velocity of ~0.32 and ~0.10 m/s forward and backward, respectively. (**b**) In SUMO1 dialyzed neurons, there was little difference in apparent conduction velocity of forward- and back-propagating action potential (~0.27 vs. 0.23 m/s, respectively). (**c**) In a representative neuron from Na$_v$1.2-Lys38Gln mutant animal, the velocity of backpropagation was not affected by SUMO1 dialysis (~0.26 vs. 0.12 m/s for forward and backpropagation, respectively). (**d**) SUMOylation causes a significant increase in the back/forward propagation velocity ratio. Each dot represents the velocity ratio obtained by measurements in individual control WT (n = 6, black), SUMO1 dialyzed WT (n = 6, blue), and SUMO1 dialyzed Na$_v$1.2-Lys38Gln mutant (n = 5, blue) axon. Box plots represent the 25–75% interquartile range of velocity ratios, and the whiskers expand to the 5–95% range. A horizontal line inside the box represents the median of the distribution, and the mean is represented by a cross symbol (X). p-Values were calculated using Student's *t*-test for unpaired data.

The online version of this article includes the following figure supplement(s) for figure 7:

**Figure supplement 1.** In computational models, SUMOylation of Na$_v$1.2 channels selectively accelerates spike backpropagation.

of somatic and axonal I$_{NaP}$ (**Fleidervish et al., 2010**; **Shvartsman et al., 2021**). Our evidence indicates that the axo-somatic difference is affected neither by Na$_v$1.2-Lys38Gln mutation nor by SUMO1 or SENP1 treatment (**Figures 4 and 5**), suggesting that some other factors confer the compartment specificity of the Na$^+$ channel gating.

The significant variability in passive and active characteristics of the WT and Na$_v$1.2-Lys38Gln mutant L5 pyramidal neurons persisted following the SUMO1 and SENP1 treatment. It is, therefore, unlikely that this variability is caused by a difference in SUMOylation, but it rather reflects the inhomogeneity of morphological and functional properties within the L5 neuronal population (**Chagnac-Amitai et al., 1990**; **Baker et al., 2018**).

We postulate that the effects of SUMO1 differ in different parts of the neuron due to the heterogeneous subcellular distribution of Na$^+$ channel subtypes and their differential susceptibility to SUMOylation. In pyramidal neurons, the SUMO1-sensitive sodium channels, Na$_v$1.2, are in the area associated with backpropagation, that is, in the soma, dendrites, and proximal parts of the AIS (**Lorincz and Nusser, 2008**; **Hu et al., 2009**; **Grubb et al., 2011**; **Plant et al., 2016**; **Liu et al., 2022**). The SUMO1-insensitive Na$_v$1.6 channels, however, are located mainly in the distal part of the AIS and in the nodes of Ranvier, that is, in the compartments associated with spike forward propagation (**Caldwell et al.,**

*2000*; *Lorincz and Nusser, 2008*; *Hu et al., 2009*; *Li et al., 2014*; *Tian et al., 2014*; *Plant et al., 2016*; *Liu et al., 2022*). The differential effects of SUMO1 on propagation speed (*Figure 7*), in addition to the differential effect of SUMO1 on the activation curve of the $I_{NaP}$ (*Figure 4*), are part of complex, compartment-specific neuromodulatory processes regulating neuronal excitability.

Our evidence that, in cortical pyramidal neurons, SUMO1 facilitates spike backpropagation but not forward propagation suggests that SUMOylation is less involved in regulating timing and synchrony in the cortical neuronal circuits. However, SUMO1, in its physiological context, may play an essential role in regulating the spike-time-dependent plasticity of dendritic spines. The backpropagating APs invading the dendrites remove $Mg^{2+}$ from NMDA receptor channels and trigger long-lasting changes in synaptic strength (*Markram et al., 1997*; *Sjöström et al., 2001*; *Holtmaat and Svoboda, 2009*). The activation of $5\text{-HT}_{1A}$ receptors decreases the success rate of AP backpropagation and enhances the segregation of axonal and dendritic activities (*Yin et al., 2017*).

Unlike phosphorylation, SUMOylation of the target proteins is reported to depend on SUMO concentration (for review, see *Flotho and Melchior, 2013*). SUMO acts as a limiting factor for conjugation because of the abundance of enzymes responsible for SUMO attachment in the cytosol. Similarly, the concentration of SUMO-specific proteases that cleave the isopeptide bonds is a limiting factor for deSUMOylation. Thus, intracellular administration of the exogenous SUMO1 and SENP1 is capable of either saturating or emptying the SUMO-conjugation sites on the ion channels, respectively, reflecting the local concentrations of the polypeptides. However, because of the complex morphological structure of L5 pyramidal neurons, diffusion of SUMO1 and SENP1 from the somatic whole-cell pipette into the cytosol is expected to be extremely slow, with half diffusion times of several hours (*Fleidervish et al., 2008*). Therefore, a limitation of this study is that the concentration of these polypeptides is expected to be significantly lower throughout the neurons than the pipette concentration, making it difficult to predict whether SUMOylation of the $Na^+$ and $K^+$ channels has reached a steady state even after our 30 min protocols.

Our results demonstrate that SUMOylation of $Na_v1.2$ channels significantly increases the speed of AP backpropagation. The subsequent events and consequences due to the acceleration may need to be further investigated, for example, the change of the $Ca^{2+}$ transient to synaptic contacts on dendrites, the alteration of local dendritic membrane excitability, and the potential effects on other neuromodulator receptors. SUMOylation might alter the time delay between the pre- and postsynaptic APs, thereby influencing the resulting change in synaptic efficiency. Together with the synergistic effect on the excitability in cortical pyramidal neurons, our findings suggest that $Na_v1.2$ and the SUMO pathway might be a new mechanism for regulating AP and neuronal function in the brain.

# Materials and methods

**Key resources table**

| Reagent type (species) or resource | Designation | Source or reference | Identifiers | Additional information |
|---|---|---|---|---|
| Peptide, recombinant protein | Human SUMO1 | R&D Systems | UL-740 | |
| Peptide, recombinant protein | Human SENP1 | R&D Systems | E-700 | |
| Strain, strain background (*Mus musculus*) | Mouse: C57BL/6N-*Scn2a*K38QMut/+ | Biocytogen | EGE-ZY-016 | |
| Software, algorithm | NEURON 8.1 | Yale University | SCR_005393 | |

## Lead contact and materials availability

Further Information and requests for resources and reagents should be directed to the lead contacts. Materials generated through this work are available from the lead contact upon reasonable request.

## Method details

### Animals

The C57BL/6N-$Na_v1.2$-K38Q$^{Mut/+}$ mice were generated by and obtained from Biocytogen (Wakefield, MA). The $Na_v1.2$-K38Q$^{Mut/+}$ mice backcrossed against C57BL/6N for five generations. Both male and female mice were used without bias. This study was carried out at the Ben-Gurion University of the Negev in accordance with the recommendations of guidelines for the welfare of experimental animals.

Animal experiments were approved by the Institutional Animal Care and Use Committee of Ben-Gurion University.

## Generation of the Scn2a$^{K38Q}$ knock in mice model

The *Scn2a*-K38Q mutation knock-in mice were generated using a CRISPR/Cas9-based approach. Briefly, two sgRNAs were designed using the CRISPR design tool (http://www.sanger.ac.uk/) to target the region of the exon 1 of the *Scn2a* gene locus, then screened for on-target activity using a Universal CRISPR Activity Assay (UCATM, Biocytogen Pharmaceuticals Co., Ltd). The T7 promoter sequence was added to the Cas9 or sgRNA template by PCR amplification in vitro. Different concentrations of the donor vector and the purified, in vitro-transcribed Cas9 mRNA and sgRNA were mixed and co-injected into the cytoplasm of one-cell stage-fertilized egg from a C57BL/6N mouse. The injected zygotes were transferred into the oviducts of Kunming pesudopregnant females to generate F0 mice. PCR and sequencing verified founder pups harboring the intended mutation were then crossed with wild-type mice for germline transmission. The germline sequence was confirmed by PCR, sequencing, and Southern blot analysis.

## Slice preparation and whole-cell recording

Experiments were performed on L5 pyramidal neurons in 300-µm-thick mouse cortical sagittal slices using previously described techniques (*Fleidervish et al., 2010*; *Katz et al., 2018*). The P18-P24 mice of either sex (Envigo, Israel) were anesthetized with isoflurane, decapitated, and the brains were placed in cold (4–8°C) oxygenated (95% $O_2$–5% $CO_2$) artificial cerebrospinal fluid (aCSF). The composition of the aCSF was (in mM) 124 NaCl, 3 KCl, 2 $CaCl_2$, 2 $MgSO_4$, 1.25 $NaH_2PO_4$, 26 $NaHCO_3$, and 10 glucose (all chemicals obtained from Sigma-Aldrich); pH was 7.4 when bubbled with 95% $O_2/CO_2$. Slices were cut on a vibratome (VT1200, Leica) and placed in a holding chamber containing oxygenated aCSF at room temperature; they were transferred to a recording chamber after at least 30 min of incubation.

The cells were viewed with a 40 or 60× water-immersion lens in a BX51WI microscope (Olympus) mounted on an X–Y translation stage (Luigs and Neumann, Germany). Somatic whole-cell recordings were made using patch pipettes pulled from thick-walled borosilicate glass capillaries (1.5 mm outer diameter; Science Products, Germany). The pipette solution for whole-cell voltage-clamp experiments contained (in mM) 135 CsCl, 2 $MgCl_2$, 4 NaCl, 10 HEPES, pH adjusted to 7.3 with CsOH (all chemicals obtained from Sigma-Aldrich) and it was supplemented with 2 mM of $Na^+$-sensitive dye, SBFI tetra-ammonium salt (Thermo Fisher Scientific) (*Minta and Tsien, 1989*). When filled with this solution, pipettes had resistance of 3–6 MΩ. Voltage-clamp recordings from L5 neurons visually identified using IR-DIC optics (*Stuart et al., 1993*) were made with a MultiClamp 700B amplifier equipped with CV-7B headstage (Molecular Devices). Data were low-pass-filtered at 2 kHz (−3 dB, 4-pole Bessel filter) and digitized at 10 kHz using Digidata 1322A digitizer driven by PClamp 9 software (Molecular Devices). Care was taken to maintain the access resistance as low as possible (usually 6–7 MΩ and always less than 10 MΩ); series resistance was 80% compensated using the built-in circuitry of the amplifier. $Ca^{2+}$ currents were blocked by adding 200 µM $Cd^{2+}$ to the bath. Voltages were not corrected for liquid junction potential. The recordings were made at room temperature (20 ± 1°C). Current-clamp recordings were made with a MultiClamp 700B amplifier (Molecular Devices). Data were low-pass-filtered at 30 kHz (−3 dB, four-pole Bessel filter) and digitized at 100 kHz. Somatic recordings were made by using patch pipettes pulled from thick-walled borosilicate glass capillaries (1.5 mm outer diameter; Hilgenberg). Pipettes had resistances of 5–7 MΩ when filled with K gluconate-based solution with the following composition (in mM): 130 K-gluconate, 6 KCl, 2 $MgCl_2$, 4 NaCl, and 10 HEPES, with pH adjusted to 7.25 with KOH. Solution was supplemented with 2 mM of sodium-binding benzofuran isophthalate (SBFI, Molecular Probes).

EPSP were elicited by delivering brief (0.1 ms) current pulses using optically coupled ISO-Flex Stimulus Isolator (AMPI, Jerusalem) via the bipolar Tungsten electrode (WPI, 0.01 MΩ) placed in the vicinity of the postsynaptic neuron. The stimulation intensity was carefully controlled to elicit monosynaptic, subthreshold EPSPs with a latency of <1ms post-stimulus.

SUMO (1 nM) and SENP (0.25 nM) were delivered to the neurons intracellularly via the whole-cell recording pipette.

## Measuring propagation speed

To measure AP propagation velocity, we performed simultaneous recordings from soma and axon of L5 pyramidal neurons. The whole-cell current-clamp somatic recordings were obtained, and the neurons were filled for 15 min with Na$^+$ indicator, SBFI (2 mM), as described above. Trains of five APs were elicited by delivering brief current steps via the somatic pipette, and axons were identified by their characteristic Na$^+$ signals. Another pipette filled with the extracellular solution supplemented with SBFI (2 mM), with a resistance of 15–20 MΩ, was positioned at different points along the axon in a loose-patch configuration. At each point along the axon, 100 single APs were elicited by delivering brief current pulses via the somatic electrode, and axonal action currents were simultaneously recorded. Both pipettes were coated within ~100 μm of the tip with Parafilm (Sigma-Aldrich) to minimize stray capacitance. Currents were low-pass-filtered at 100 kHz (−3 dB, four-pole Bessel filter) and digitized at 200 kHz. To identify the time delays between the somatic and the axonal signals, they were aligned to the times of maximal rate of rise of the somatic APs and averaged. Then, the differences between the times of peak of the axonal action currents and times of maximal rate of rise of the somatic APs were calculated.

## Sodium imaging

Imaging experiments were performed as described previously (*Baranauskas et al., 2013*; *Shvartsman et al., 2021*). SBFI fluorescence was excited by using a high-intensity LED device (385 ± 4 nm; Prizmatix), and the emission was collected by using a modified Olympus U-MNU2 filter set (400 nm dichroic mirror; 420 nm long-pass emission filter). The fluorescent response of SBFI was recorded using a back-illuminated 80 × 80 pixel cooled camera (NeuroCCDSMQ; RedShirt Imaging) at 500 frames/s acquisition speed controlled by Neuroplex software. Indicator bleaching was corrected by subtracting an equivalent blank trace without electrical stimulation.

## Data analysis

Data analysis was accomplished using pCLAMP10 software (Molecular Devices) and Origin 6.0 (OriginLab). If not otherwise noted, values are given as mean ± SE. Student's *t* test was used for statistical analysis.

## Modeling

Numerical simulations were performed in the NEURON simulation environment (*Hines and Carnevale, 1997*). Unless otherwise stated, electrophysiological parameters and dynamic [Na$^+$]$_i$ changes were studied in a simplified compartmental model encompassing the fundamental morphological and electrical features of layer 5 pyramidal neurons as described previously (*Baranauskas et al., 2013*; *Shvartsman et al., 2021*).

In the model, the 1.2-μm-thick AIS extended over the first 40–50 μm of the axon. The subsequent segment (length, 50 μm; diameter, 1.2 μm) was myelinated. The nodes were 1 μm long and had a diameter of 1.2 μm, and the myelinated internodes were 2 μm long and had a diameter of 1.2 μm. In addition to the axon, the soma (length 35 $\mu m$, diameter: 23 $\mu m$) gave rise to the apical dendrite (length 350 $\mu m$, diameter 3.5 $\mu m$) and two basal dendrites (length 200 $\mu m$, diameter 1.2 $\mu m$). For spatial precision, all compartments were divided into 1-μm-long isopotential segments.

The passive electrical properties R$_m$, C$_m$, and R$_i$ were set to 25,000 Ω cm$^2$, 1 μF cm$^{-2}$, and 150 Ω cm, respectively, uniformly. The myelinated internode had C$_m$ of 0.5 μF·cm$^{-2}$. The resting membrane potential at the soma was set to −75 mV.

All simulations were run with 1-μs time steps, and the nominal temperature was set to 18°C. The model used a Hodgkin–Huxley-based Na$^+$ conductance. The steady-state activation and inactivation characteristics of the Na$_v$1.6 channels were left-shifted by 6 mV and 3 mV, respectively, compared with the Na$_v$1.2 channels. The Na$^+$ conductance was set to 200 pS μm$^{-2}$ in the soma, 200 pS μm$^{-2}$ in the apical dendrite, 40 pS μm$^{-2}$ in the basal dendrites, 1200 pS μm$^{-2}$ in the nodes of Ranvier; no Na$^+$ channels were present in the internodes. The model included Kv and Kv1 K$^+$ channels with kinetics and density as previously described. The K$^+$ equilibrium potential was set to −85 mV.

The AIS contained variable Na$^+$ channel density as described by *Baranauskas et al., 2013*. At both proximal and medial parts of the AIS, the gNa was represented only by Na$_v$1.2 channels. The gNa at

the proximal AIS segment incremented linearly from 200 pS µm$^{-2}$ to 800 pS µm$^{-2}$, the middle AIS part had a constant gNa of 800 pS µm$^{-2}$. The distal AIS part was populated by Na$_v$1.6 channels with density decrementing from 800 to 0 pS µm$^{-2}$.

Diffusion of Na$^+$ ions was modeled as the exchange of Na$^+$ ions between adjacent neuronal compartments using the intrinsic protocols in NEURON, assuming a diffusion coefficient of 0.6 µm$^2$ ms$^{-1}$ (*Kushmerick and Podolsky, 1969*; *Fleidervish et al., 2010*). The resting intracellular and the extracellular Na$^+$ concentrations were set to 4 and 151 mmol/l, respectively.

## Acknowledgements

This research was supported by the Israel Science Foundation (grant no. 1384/19) and National Institutes of Health grant R01HL10549 (to SANG).

## Additional information

### Funding

| Funder | Grant reference number | Author |
| --- | --- | --- |
| Israel Science Foundation | 1384/19 | Ilya Fleidervish |
| National Institutes of Health | R01HL10549 | Steven AN Goldstein |

The funders had no role in study design, data collection and interpretation, or the decision to submit the work for publication.

### Author contributions

Oron Kotler, Leigh D Plant, Data curation, Formal analysis, Investigation, Writing - original draft, Writing - review and editing; Yana Khrapunsky, Data curation, Formal analysis, Investigation, Project administration; Arik Shvartsman, Hui Dai, Data curation, Formal analysis, Investigation; Steven AN Goldstein, Ilya Fleidervish, Conceptualization, Resources, Data curation, Formal analysis, Supervision, Funding acquisition, Validation, Investigation, Methodology, Writing - original draft, Project administration, Writing - review and editing

### Author ORCIDs

Oron Kotler http://orcid.org/0000-0003-1698-545X
Hui Dai http://orcid.org/0000-0002-0440-7815
Leigh D Plant http://orcid.org/0000-0002-1622-1655
Steven AN Goldstein http://orcid.org/0000-0001-5207-5061
Ilya Fleidervish http://orcid.org/0000-0002-5501-726X

### Ethics

This study was carried out at the Ben-Gurion University of the Negev in accordance with the recommendations of guidelines for the welfare of experimental animals. Animal experiments were approved by the Institutional Animal Care and Use Committee of Ben-Gurion University (protocols IL-68-09-2019(A), IL-79-10-2020(D)).

### Decision letter and Author response

Decision letter https://doi.org/10.7554/eLife.81463.sa1
Author response https://doi.org/10.7554/eLife.81463.sa2

## Additional files

### Supplementary files

• MDAR checklist

## Data availability

All data generated or analyzed during this study are included in the manuscript and supporting file; the Source Data files are uploaded to Dryad.

The following dataset was generated:

| Author(s) | Year | Dataset title | Dataset URL | Database and Identifier |
|---|---|---|---|---|
| Fleidervish I, Kotler O, Khrapunsky Y, Shvartsman A, Dai H, Plant L, Goldstein S | 2023 | SUMOylation of NaV1.2 channels regulates the velocity of backpropagating action potentials in cortical pyramidal neurons | https://dx.doi.org/10.5061/dryad.tx95x6b1g | Dryad Digital Repository, 10.5061/dryad.tx95x6b1g |

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
