## [Editor Report]

This fundamental study describes how the specific SUMOylation of Nav1.2 channels regulates neuronal function by slowing action potential backpropagation from the AIS. This compelling evidence breaks new ground in the role of SUMOylation in modulating synaptic plasticity and will be of interest to neuroscientists working on synaptic transmission and modulation of ion channel activity.

---

## [Decision Letter]

**Decision letter after peer review:**

Thank you for submitting your article "SUMOylation of NaV1.2 channels regulates the velocity of backpropagating action potentials in cortical pyramidal neurons" for consideration by *eLife*. Your article has been reviewed by 3 peer reviewers, and the evaluation has been overseen by a Reviewing Editor and John Huguenard as the Senior Editor. The following individual involved in review of your submission has agreed to reveal their identity: Theodore R Cummins (Reviewer #3).

Essential revisions:

All reviewers agreed that these results are solid and interesting. However, reviewers also raised several concerns about the interpretation of the data and some other aspects related to data analysis and discussion that should be addressed by the authors. Essential revisions should include:

1) Discuss the effect of SUMOylation on the frequency of firing.

2) Review statistical analysis and/or data in all Figures, following the suggestions by reviewer #2.

3) Discuss potential differences in neuronal types used in the study, and explain differences in neuronal parameters after treatments identified in some experiments (e.g. Figure 1-Sup. 1).

4) Provide an extended comparison of Nav1.2-Lys38Gln with wild type neuronal properties in similar experimental conditions.

5) Discuss the limitations of the slice recordings and alternative interpretations of the obtained results using ramp-elicited currents, e.g. what/if other sodium current properties, not examined in this study, could contribute to the observed results. Additionally, discuss how this interpretation relates to the changes observed in the previous studies (e.g., Plant et al., 2016).

6) Revise the text according to all recommendations raised by the reviewers and listed in the individual reviews below.

Because the individual reviews include several important points they are included here for your reference.

*Reviewer #1 (Recommendations for the authors):*

It seemed odd that the effects of SUMO1 and SENP1 on the f-I curve in wild-type mice were relegated to a Supplemental figure while the lack of effect on the f-I slope in the engineered mouse was in a main figure – it seemed that the opposite (or showing both in a main figure) would be more logical.

322 "We observe that the effects of SUMO1 differs in different parts of"…should be "differ" to match "effects".

338 "Our evidence that SUMO1 affects the backpropagation, but not forward propagation indicates that activation of SUMOylation pathways does not affect the ongoing processing by the cortical neuronal circuits."

It seems that the dramatic changes in the f-I relation do constitute a significant change in processing.

232 and 579 "leftward shift of activation kinetics" might be more precisely "leftward shift of voltage-dependence", since it is the voltage-dependence rather than kinetics that is shown.

*Reviewer #2 (Recommendations for the authors):*

Regarding the point 1 from "public review", I would like to mention two examples for better understanding:

– Lines 156-158: "mean instantaneous firing frequency in response to a 0.3 nA current injection increased from 14.7{plus minus}3.9 Hz immediately after the break-in to 25.7{plus minus}5.4 Hz (n=6, p=0.029) after 35 mins of SUMO1 dialysis."

Values are mean +/- S.E. Doing the statistics (GraphPad, InStat, Statgraphics, etc.), using Student's t-test, we obtain that the value of t = 1.651 (with 10 degrees of freedom) and P = 0.1297 (not significant).

– Lines 168-169, input resistance values: "from 96.8 {plus minus} 12.9 MΩ at a time of break-in to the cell to 120.7 {plus minus} 14.5 MΩ (n=6, p<0.03) at 35 min of recording."

Doing the statistics, the value of t = 1.231 (with 10 degrees of freedom) and P = 0.2463 (not significant).

These are just two examples, but all the data should be thoroughly revised. I agree that the trends are perceived in several cases, but, unfortunately, with those data values and number of replications, statistics shown not significant differences in most of the cases.

*Reviewer #3 (Recommendations for the authors):*

Very interesting study. I am concerned about the statement that "exclusively controls InaP generation" in the abstract and the focus on persistent current in the text. The previously described leftward shift in voltage-dependence of activation for peak sodium currents likely would contribute to the observed changes in ramp-elicited currents. It is possible that the voltage-dependence of activation is also altered in the cortical neurons. The manuscript should better discuss the limitations of the slice recordings and how the changes observed in the previous studies (e.g., Plant et al., 2016) are not invalidated by the data presented in the current study.

The paper also is deficient in details on the "SUMO1 or SENP1 peptides".

It seems that the whole SUMO1 protein is used. While some might call a 101 amino acid polymer a peptide, if the full length protein is used, protein is probably more accurate. SENP1, on the other hand, does not refer to the full-length protein. It refers to a recombinant protein that seems to be 229 amino-acids and contains the catalytic subunit of SENP1. It would be good to clarify what is being used and tighten up the language so that it is accurate.

Is there a change in resting membrane potential with the recombinant proteins? Is input resistance estimated from the resting membrane potential? These details may be important and should be included.

---

## [Author Response]

Essential revisions:1) Discuss the effect of SUMOylation on the frequency of firing.

The Discussion on the effect of SUMOylation on the frequency of firing has been extended, and the relevant paper by Chance, Abbott and Reyes (2002) has been cited (see Discussion, p. 14).

2) Review statistical analysis and/or data in all Figures, following the suggestions by reviewer #2.

Done. We thank Reviewer #2 for checking our statistical analysis. His calculations using Student's t-test would be valid for comparing two independent (unpaired) datasets. Our evidence, however, is mostly based on paired data analysis: the parameters obtained immediately after the breakin to whole-cell configuration were compared to those measured after drug diffusion into the cell. We now mention that Student's t-test for paired data was used for statistics in Figure Legends, where appropriate.

We also added more data points for most experiments. The data are now presented as box plots representing the 25–75% interquartile range, with the whiskers expanding to the 5–95% range. A horizontal line inside the box represents the median of the distribution, and the mean is represented by a cross symbol (X). The paired values obtained from the same individual neuron are presented as dots connected by a line.

3) Discuss potential differences in neuronal types used in the study, and explain differences in neuronal parameters after treatments identified in some experiments (e.g. Figure 1-Sup. 1).

Done, see Discussion, p. 15

4) Provide an extended comparison of Nav1.2-Lys38Gln with wild type neuronal properties in similar experimental conditions.

Table 1 comparing the passive and active parameters of Nav1.2-Lys38Gln and WT neurons has been added. Results on p. 9 has also been modified accordingly.

5) Discuss the limitations of the slice recordings and alternative interpretations of the obtained results using ramp-elicited currents, e.g. what/if other sodium current properties, not examined in this study, could contribute to the observed results. Additionally, discuss how this interpretation relates to the changes observed in the previous studies (e.g., Plant et al., 2016).

The Discussion has been added, see p. 14.

6) Revise the text according to all recommendations raised by the reviewers and listed in the individual reviews below.

Done, see below.

Reviewer #1 (Recommendations for the authors):It seemed odd that the effects of SUMO1 and SENP1 on the f-I curve in wild-type mice were relegated to a Supplemental figure while the lack of effect on the f-I slope in the engineered mouse was in a main figure – it seemed that the opposite (or showing both in a main figure) would be more logical.

We revised the Figures in accordance with Reviewer #1 suggestion: the effects of SUMO1 and SENP1 on the F-I curve in wild-type mice are now presented as Figure 2.

322 "We observe that the effects of SUMO1 differs in different parts of"…should be "differ" to match "effects".

Corrected.

338 "Our evidence that SUMO1 affects the backpropagation, but not forward propagation indicates that activation of SUMOylation pathways does not affect the ongoing processing by the cortical neuronal circuits."It seems that the dramatic changes in the f-I relation do constitute a significant change in processing.

This sentence is now corrected as follows:

“Our evidence that SUMO1 facilitates spike backpropagation but not forward propagation indicates that SUMOylation pathways are probably less involved in modifying timing and synchrony in the cortical neuronal circuits”.

232 and 579 "leftward shift of activation kinetics" might be more precisely "leftward shift of voltage-dependence", since it is the voltage-dependence rather than kinetics that is shown.

Corrected throughout the text.

Reviewer #2 (Recommendations for the authors):Regarding the point 1 from "public review", I would like to mention two examples for better understanding:– Lines 156-158: "mean instantaneous firing frequency in response to a 0.3 nA current injection increased from 14.7{plus minus}3.9 Hz immediately after the break-in to 25.7{plus minus}5.4 Hz (n=6, p=0.029) after 35 mins of SUMO1 dialysis."Values are mean +/- S.E. Doing the statistics (GraphPad, InStat, Statgraphics, etc.), using Student's t-test, we obtain that the value of t = 1.651 (with 10 degrees of freedom) and P = 0.1297 (not significant).– Lines 168-169, input resistance values: "from 96.8 {plus minus} 12.9 MΩ at a time of break-in to the cell to 120.7 {plus minus} 14.5 MΩ (n=6, p<0.03) at 35 min of recording."Doing the statistics, the value of t = 1.231 (with 10 degrees of freedom) and P = 0.2463 (not significant).These are just two examples, but all the data should be thoroughly revised. I agree that the trends are perceived in several cases, but, unfortunately, with those data values and number of replications, statistics shown not significant differences in most of the cases.

We thank Reviewer #2 for the careful and thorough examination of our data and Figures. As we pointed out above, however, our measurements are paired, i.e., taken before and after treating the same individual neuron with SUMO1, SENP, or control solution. Therefore, using a paired t-test is justified. We now mention that Student's t-test for paired data was used for statistics in Figure Legends, where appropriate.

Following the Reviewer’s request, we carefully inspected our datasets and added more data points for most experiments.

Reviewer #3 (Recommendations for the authors):Very interesting study. I am concerned about the statement that "exclusively controls InaP generation" in the abstract and the focus on persistent current in the text. The previously described leftward shift in voltage-dependence of activation for peak sodium currents likely would contribute to the observed changes in ramp-elicited currents. It is possible that the voltage-dependence of activation is also altered in the cortical neurons. The manuscript should better discuss the limitations of the slice recordings and how the changes observed in the previous studies (e.g., Plant et al., 2016) are not invalidated by the data presented in the current study.

We have corrected this misleading sentence in the abstract.

We focused on I_NaP_ and not on transient Na^+^ current because it is impossible to voltage-clamp large spatially distributed pyramidal neurons in slices. The findings by Plant et al., 2016 are not invalidated by the current study. Indeed, our evidence that SUMOylation/deSUMOylation shifts I_NaP_ activation indirectly supports Plant et al., 2016 findings, although we cannot confirm that the effects on transient and persistent Na^+^ currents are identical.

We added a paragraph to the Discussion that explains these issues.

The paper also is deficient in details on the "SUMO1 or SENP1 peptides".It seems that the whole SUMO1 protein is used. While some might call a 101 amino acid polymer a peptide, if the full length protein is used, protein is probably more accurate. SENP1, on the other hand, does not refer to the full-length protein. It refers to a recombinant protein that seems to be 229 amino-acids and contains the catalytic subunit of SENP1. It would be good to clarify what is being used and tighten up the language so that it is accurate.

Done.

Is there a change in resting membrane potential with the recombinant proteins?

We failed to detect any consistent, significant change in the resting membrane potential with SUMO1 and SENP1.

Is input resistance estimated from the resting membrane potential?

Yes, we added the sentence to the Methods.